# A highly-contiguous genome assembly of the Eurasian spruce bark beetle, *Ips typographus*, provides insight into a major forest pest

Daniel Powell [1,2,7], Ewald Große-Wilde[1], Paal Krokene[3], Amit Roy[1], Amrita Chakraborty[4], Christer Löfstedt [2], Heiko Vogel[5], Martin N. Andersson [2,8 ✉] & Fredrik Schlyter [1,6,8]

Conifer-feeding bark beetles are important herbivores and decomposers in forest ecosystems. These species complete their life cycle in nutritionally poor substrates and some can kill enormous numbers of trees during population outbreaks. The Eurasian spruce bark beetle (*Ips typographus*) can destroy >100 million $m^3$ of spruce in a single year. We report a 236.8 Mb *I. typographus* genome assembly using PacBio long-read sequencing. The final phased assembly has a contig $N_{50}$ of 6.65 Mb in 272 contigs and is predicted to contain 23,923 protein-coding genes. We reveal expanded gene families associated with plant cell wall degradation, including pectinases, aspartyl proteases, and glycosyl hydrolases. This genome sequence from the genus *Ips* provides timely resources to address questions about the evolutionary biology of the true weevils (Curculionidae), one of the most species-rich animal families. In forests of today, increasingly stressed by global warming, this draft genome may assist in developing pest control strategies to mitigate outbreaks.

[1] Czech University of Life Sciences Prague, Faculty of Forestry and Wood Sciences, Excellent Team for Mitigation (ETM), Kamýcká 129, Praha 6, Suchdol, Czech Republic. [2] Department of Biology, Lund University, Lund, Sweden. [3] Division of Biotechnology and Plant Health, Norwegian Institute of Bioeconomy Research, Ås, Norway. [4] Czech University of Life Sciences Prague, Faculty of Forestry and Wood Sciences, EVA 4.0 Unit, Kamýcká 129, Praha 6, Suchdol, Czech Republic. [5] Entomology Department, Max Planck Institute for Chemical Ecology, Jena, Germany. [6] Department of Plant Protection Biology, Swedish University of Agricultural Sciences, Alnarp, Sweden. [7] Present address: Global Change Ecology Research Group, School of Science, Technology and Engineering, University of the Sunshine Coast, Sippy Downs, QLD, Australia. [8] These authors contributed equally: Martin N. Andersson, Fredrik Schlyter. ✉email: martin_n.andersson@biol.lu.se

Conifer-feeding bark beetles (Coleoptera; Curculionidae; Scolytinae) are keystone species in forest ecosystems, contributing to wood decomposition and nutrient recycling through direct feeding and the action of beetle-associated microbiota[1]. However, a few so-called aggressive species can also kill large numbers of healthy trees through pheromone-coordinated mass-attacks once their populations surpass a critical threshold density, quickly transforming entire forest landscapes at tremendous economic and ecological costs[2,3]. Since abiotic factors are important drivers for beetle population growth[4], outbreaks of aggressive species are expected to increase in frequency and severity due to climate change[5,6]. Beetle population growth is promoted by increasing temperatures that reduce developmental time, allowing for the production of additional generations per year. Furthermore, the resistance of conifer trees to beetle attack is compromised during heat and drought stress[7].

The Eurasian spruce bark beetle (*Ips typographus* [L.]) is the most serious pest of Norway spruce (*Picea abies* [L.] Karst) and other spruce species in the beetle's range, currently causing unprecedented forest destruction across the Palearctic region (Fig. 1). In Europe, *I. typographus* killed between 2 and 14 million m³ of spruce annually from 1970 to 2010[8], but during the hot and dry summer of 2019, it killed more than 118 million m³ [9]. Beetle mass-attacks on trees are coordinated by an aggregation pheromone, which is produced by males as they initiate boring in suitable host trees and attracts large numbers of both sexes[10]. Pheromone attraction is modulated by other compounds produced by con- and hetero-specific beetles as well as by host and non-host trees[11–13]. Due to the importance of volatile signals in the ecology of *I. typographus*, transcriptomes from antennal tissue have been generated to address the molecular biology and function of chemosensation[14–16]. However, genetic resources are otherwise scarce for this species, and

such resources are needed to better understand the specialised ecology of *I. typographus*, and ultimately to develop improved pest management strategies.

Tree-killing bark beetles like *I. typographus* thrive on the inner bark (Fig. 1a–d), one of the most nutritionally poor and recalcitrant organic substrates on earth[17]. Abundant carbon-rich bio-polymers present in bark and adjoining sapwood, such as lignin, hemicellulose and cellulose, may not be available to bark beetles as nutrients until prior degradation by fungi and other microbes[18]. Bark beetles, including *I. typographus*, are associated with a diverse microbiota, including gut endo-symbionts[19,20], yeasts[21], and ecto-symbionts such as ophiostomatoid fungi that the beetles inoculate into attacked trees[22]. Once established in the tree, these fungi may serve as food for developing beetles, metabolise conifer chemical defences, and accelerate tree death[23–25].

Although beetles (Coleoptera) constitute the largest order of the Metazoa, only 11 Coleoptera genomes have been published, of which only two belong to members of the 6,000 species strong Scolytinae subfamily (the mountain pine beetle, *Dendroctonus ponderosae* Hopkins[26], and the coffee berry borer, *Hypothenemus hampei* Ferrari[27]). Sequencing of additional Scolytinae genomes would enable comparative genomic studies and shed light on taxon-specific ecological adaptations in this highly specialized and important insect group. The main aim of this study was to assemble and annotate a high-quality genome of *I. typographus*. Secondly, we performed a comparative genomic analysis with 20 other metazoan genomes, specifically investigating *Ips*- and bark beetle-specific gene family expansions. We expect that our high-quality assembly of the *I. typographus* genome will be an important resource for forthcoming fundamental and applied studies on this important insect that endangers entire forest landscapes (Fig. 1e).

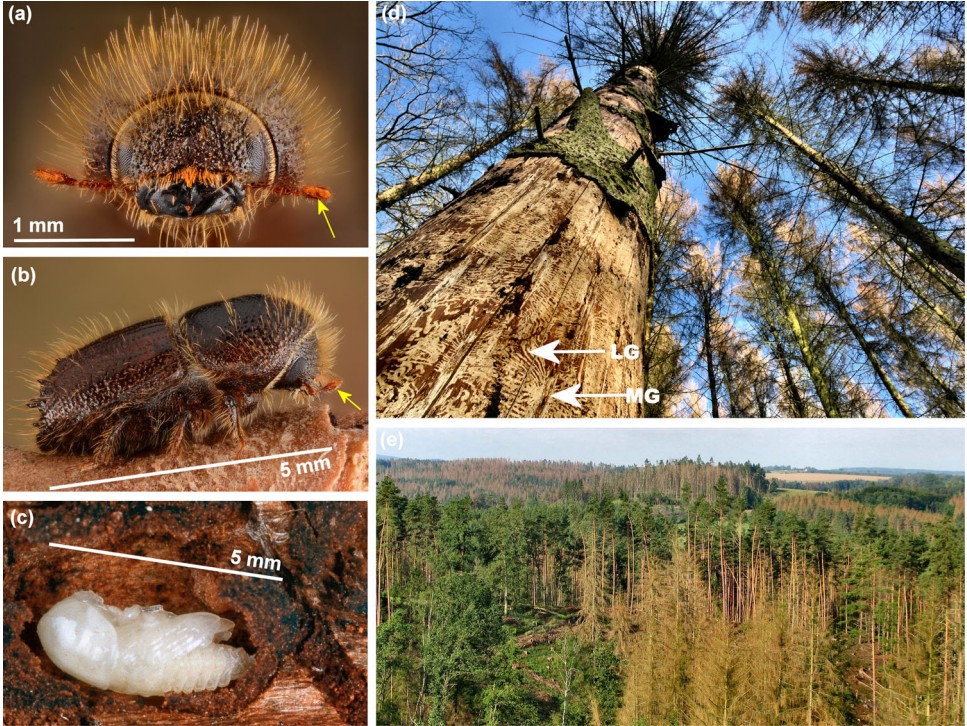

**Fig. 1 The Eurasian spruce bark beetle.** Adult *Ips typographus* (**a**) front and (**b**) side view with sensory array on antennae indicated (yellow arrows). **c** Pupae in bark. **d** Damage on Norway spruce (*Picea abies*) at tree level (with one of several hundred galleries indicated with white arrows; MG = straight egg tunnel made by one mother and LG = winding larval tunnels), **e** Damage at landscape level of mixed forest with Norway spruce (all brown), while trees with green crowns are Scots pine (*Pinus sylvestris*) or deciduous trees. Pictures (**a**, **b**, **d**) from Belgium by Gilles San Martin (URL: https://flickr.com/people/sanmartin/), under CC BY-SA 2.0, (**c**) by blickwinkel, Alamy Stock Photo (licenced), and (**e**) from Czech Republic, Central Bohemia, Koberovice, August 2019 by Jan Liška.

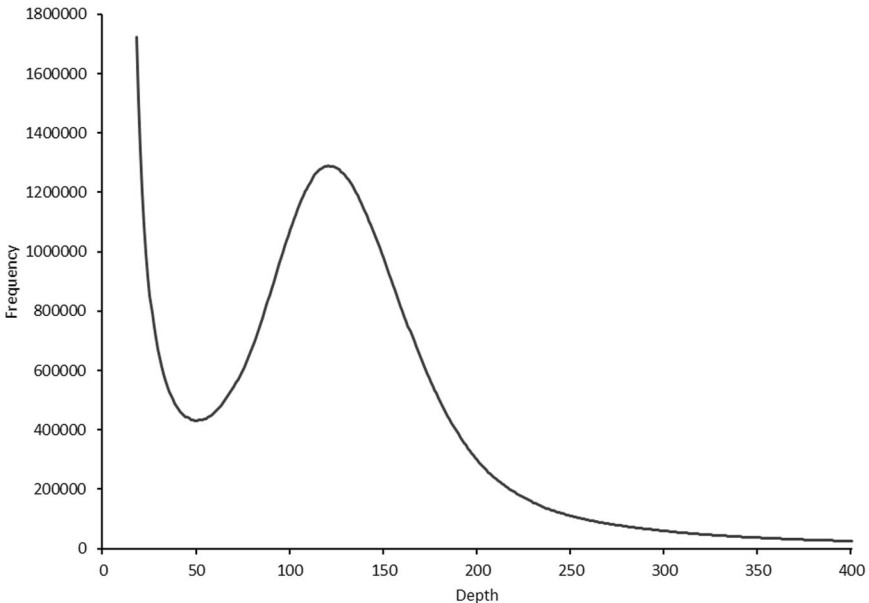

**Fig. 2 K-mer distribution of Illumina sequencing reads of Ips typographus.** The peak $k$-mer depth ($k = 19$) was 124, giving an estimated haploid genome length of 221,929,798 bp. The presence of a single peak suggests a highly homozygous genome.

**Table 1 Assembly statistics of the *Ips typographus* genome. Comparisons with *I. typographus* and two other Coleoptera species, *Dendroctonus ponderosae* and *Tribolium castaneum*.**

| Statistic | Ityp1.0 (this study) | DendPond_male_1.0[a] | Tcas5.2[b] |
|---|---|---|---|
| Genome size (Mb) | 236.82 | 252.85 | 165.93 |
| Number of contigs | 272 | 8,188 | 2,081 |
| Contig N50 (Mb) | 6.65 | 0.63 | 15.27 |
| Contig L50 | 12 | 87 | 5 |
| Max. contig length (Mb) | 16.86 | 4.16 | 31.38 |
| Min. contig length (kb) | 20.29 | 1.00 | 0.25 |
| GC (%) | 35.21 | 35.91 | 33.86 |
| BUSCO Completeness (insecta_odb9) | 98.6% | 95.4% | 99.3% |
| Complete and single-copy | 1,535 (92.6%) | 1,489 (89.8%) | 1,643 (99.1%) |
| Complete and duplicated | 99 (6%) | 93 (5.6%) | 4 (0.2%) |
| Fragmented | 15 (0.9%) | 49 (3.0%) | 8 (0.5%) |
| Missing | 9 (0.5%) | 27 (1.6%) | 3 (0.2%) |
| RNA-Seq alignment rate (% total) | 93.56 | - | - |
| RNA-Seq alignment rate (% concordantly exactly 1 time) | 83.73 | - | - |

Note: Comparison genomes downloaded from GenBank; [a]*Dendroctonus ponderosae* GCA_000355655.1; [b]*Tribolium castaneum* GCA_000002335.3.

## Results and discussion

**Genome sequencing, assembly, and completeness assessment.** A highly inbred line of *I. typographus* was successfully produced via full-sib mating for 10 generations. Whole-genome sequencing yielded over 4.5 million subreads totalling more than 50 gigabases of sequence equivalent to 211-fold coverage. The subread $N_{50}$ was 19.3 kb with a mean read length of 12.4 kb (Supplementary Fig. 1). Estimation of genome size using a $k$-mer-counting approach suggested a total haploid size of 221.9 Mb (Fig. 2), comparable with the mountain pine beetle (*D. ponderosae*) genome assembly of 204 Mb[26]. Flow cytometry has been performed on *I. typographus* previously to estimate genome size using both male and female beetles ($n = 6$). The result equates to an average genome size of 259 and 273 Mb for males and females, respectively, corroborating the results from our $k$-mer analysis (personal communication, J. Spencer Johnston and Anthony Cognato). The karyotype of *I. typographus* is $14 + Xy_p$[28] with the male being the heterogametic sex. Our final, phased genome sequence was 236.8 Mb in total length and comprised 272 contigs with an $N_{50}$

of 6.65 Mb; the longest contig being 16.9 Mb in size (Table 1). Seventy-eight percent of the genome assembly was contained in 36 contigs that were all greater than 1 Mb in size (Supplementary Table 1). The five largest contigs were all greater than 10 Mb in length.

Telomeric motifs are regions of repetitive sequences of DNA that indicate the ends of chromosomes. These motifs were identified at the ends of eight different contigs (five forward strands, three reverse strands). Telomeric motifs were located at one end of the five largest contigs (Supplementary Table 2). The overall GC content was 35.21%. Approximately 28.2% of the genome contained repetitive sequences when masked using a custom library (Supplementary Table 3). This was slightly higher than in *D. ponderosae* (between 17% and 23%)[26], though lower than in *Tribolium castaneum* (42%)[29]. Analysis of completeness of the draft genome using the BUSCO tool revealed that 1,649 (99.5%) of the 1,658 genes in the insecta_odb9 database could be identified as present either partial or complete. Only nine genes (0.5%) were considered missing from the assembly. The *I.*

**Table 2 Statistics of the predicted gene models from the _Ips typographus_ draft assembly (Ityp1.0).**

| Statistic | Ityp1.0 |
|---|---|
| Number of genes | 23,923 |
| Total gene length (bp) | 132,911,182 |
| Longest gene (bp) | 318,767 |
| Average gene length (bp) | 5,556 |
| Average exon length (bp) | 324 |
| Average intron length (bp) | 957 |
| % of genome covered by genes | 56 |
| Average exons per gene | 5 |
| Average introns per gene | 4 |
| Number containing Pfam domains | 14,145 |

_typographus_ assembly statistics were of a quality comparable with the release of the enhanced _T. castaneum_ genome sequence[30] and were considerably higher than most other Coleoptera genomes published to date. Thus, our evaluation indicates that the de novo assembly of the _I. typographus_ genome is of high quality. Comparisons of contig sizes with other species and the presence of multiple telomeric motifs suggest some of the largest contigs contained in this assembly are approaching chromosome scale.

**Genome annotation**. Our approach to annotate the _I. typographus_ genome exploited extensive transcriptomic resources and numerous publicly available protein sequences using ab initio and homology-based predictors. The final set of models consisted of 23,923 protein-coding genes (Table 2, Supplementary Table 4) with the majority (> 77%) assigned an annotation edit distance (AED) score of 0.5 or less (Supplementary Fig. 2). At least 84% (20,094) of the gene models were supported by alignments to our RNA-Seq read data. Searches of the annotated predicted protein sequences using BUSCOs from the insecta_odb9 libraries identified 1,584 (95.5%) of the 1,658 protein sequences in the Insecta dataset, with only 38 (2.3%) sequences missing (Supplementary Figure 3). Comparing the outcome of BUSCO searches with the published Scolytinae (_D. ponderosae_ and _H. hampei_) gene models, the _I. typographus_ predictions had similar levels of completeness as these and were also comparable with the higher-quality coleopteran genome assemblies, such as the two other wood-feeding beetles (_Agrilus planipennis_ and _Anoplophora glabripennis_; Supplementary Fig. 3). Moreover, read alignment of RNA-Seq data to annotated regions of the draft genome resulted in an average of 83.7% of reads mapping concordantly in pairs to the gene space. An overall average of 93.6% mapped reads indicates that the majority of transcripts captured in our RNA-Seq data are represented in the _I. typographus_ gene set. Of the 23,923 predicted protein sequences, 59% contained Pfam domains. The number of gene models for _I. typographus_ was considerably higher than that reported for _D. ponderosae_ (13,088)[26], though comparable with the annotation of _A. glabripennis_[31] and _H. hampei_[27] with 22,035 and 19,222 predicted protein-coding genes, respectively. Taken together, these statistics suggest the production of a comprehensive set of gene models for _I. typographus_.

**Orthology and comparative analysis with other Coleopteran genomes**. Comparison of orthologous genes shared among three beetle species closely related to _I. typographus_, two bark beetles and the polyphagous cerambycid wood-borer _A. glabripennis_, showed a core set of 6,991 shared gene clusters and a unique set of 811 clusters specific to _I. typographus_ (Fig. 3a). Gene ontology enrichment analysis of these _Ips_-specific gene clusters revealed nine enriched categories, including DNA and RNA binding, regulation, and signalling (Supplementary Table 5). A total of 22

gene families were significantly expanded in _I. typographus_ when compared with the 11 other published coleopteran genomes and a selection of other model invertebrates using a protein family domain analysis approach (Supplementary Table 6). A selection (17) of these expanded gene families is shown in Fig. 3b (see also Supplementary Table 7). As the other five gene families were domain model variations overlapping the reported domains, they were omitted from the figure. Using this comprehensive approach, we were able to highlight gene families that appear distinctly relevant to the ecology of _I. typographus_ and/or conifer-feeding bark beetles. A phylogenetic tree inferred from single-copy orthologous gene families shows the phylogenetic position of _I. typographus_ relative to the other coleopterans included in this study (Fig. 3c). An analysis of the orthologous gene groups using the CAFE approach detected 1,065 expanded and 2,218 contracted orthologous gene groups containing 6,980 genes. Given that the CAFE approach resulted in a large proportion of genes (almost a third of all predicted _Ips_ genes) being identified as either expanded or contracted, we discuss the expanded gene families identified using the PFam domain analysis approach to highlight gene families of interest based on function.

Feeding on the stem bark of trees requires an ability to metabolize plant cell walls using endogenous enzymes or relying on the actions of associated microbiota. Notably, genes containing domains associated with plant cell wall degrading enzymes were significantly expanded in the _I. typographus_ genome. Large expansions of plant cell wall degrading enzymes were previously reported in the bark beetle _D. ponderosae_ and the cerambycid wood-borer _A. glabripennis_[26,31]. In particular, three families of glycosyl hydrolase (GH) enzymes were expanded in the conifer-feeding bark beetles _I. typographus_ and _D. ponderosae_ (Fig. 3b). Genes containing the Pfam domain GH48 (PF02011) were most abundant in these species, possessing eight (_I. typographus_) and six (_D. ponderosae_) proteins containing this domain. GH48 enzymes are reducing, end-acting cellobiohydrolases and have been identified in several polyphagous insect species, but chiefly among the two coleopteran superfamilies Chrysomeloidea and Curculionoidea[32]. A detailed phylogenetic analysis of the Coleoptera and their GH gene repertoire has been reported previously[33]. The most abundant expression of these genes in _I. typographus_ occurred predominantly in the fat body, with some genes being specifically expressed during different larval stages, but none in pupae (Fig. 4a). This finding is surprising since these genes typically are highly expressed in guts, and further experiments are needed to investigate functional roles of these proteins in the fat body. GH enzymes are also commonly found in bacteria and more rarely in fungi[34] and their presence in insects is thought to be the result of horizontal gene transfer events from bacteria[32,33,35,36]. Fungal symbionts assist _I. typographus_ in colonising spruce trees and may play essential roles in beetle nutrition and weakening host tree defences[24,25]. GH48 enzymes are thought to also assist in the degradation of fungal chitin[37], which may be important for _I. typographus_. Laboratory bioassays have shown a strong feeding preference of immature adults for media colonized by symbiotic fungi[24]. Three pairs of GH48 genes are found to occur adjacent to each other in the _I. typographus_ genome. One of these pairs (Ityp12594 and Ityp12595) appear to be the result of a tandem duplication event (Supplementary Fig. 4). Phylogenetic analysis of all the GH48 domain-containing genes in the 11 species of Coleoptera analysed in this study highlights that the eight genes from _I. typographus_ share the highest similarity with those from the other Scolytinae species _D. ponderosae_ and _H. hampei_ (Fig. 4b).

The _I. typographus_ genome contains 33 aspartyl protease (PF13650) domain-containing genes. Although these genes are also expanded in the wood-borer _A. glabripennis_, the number

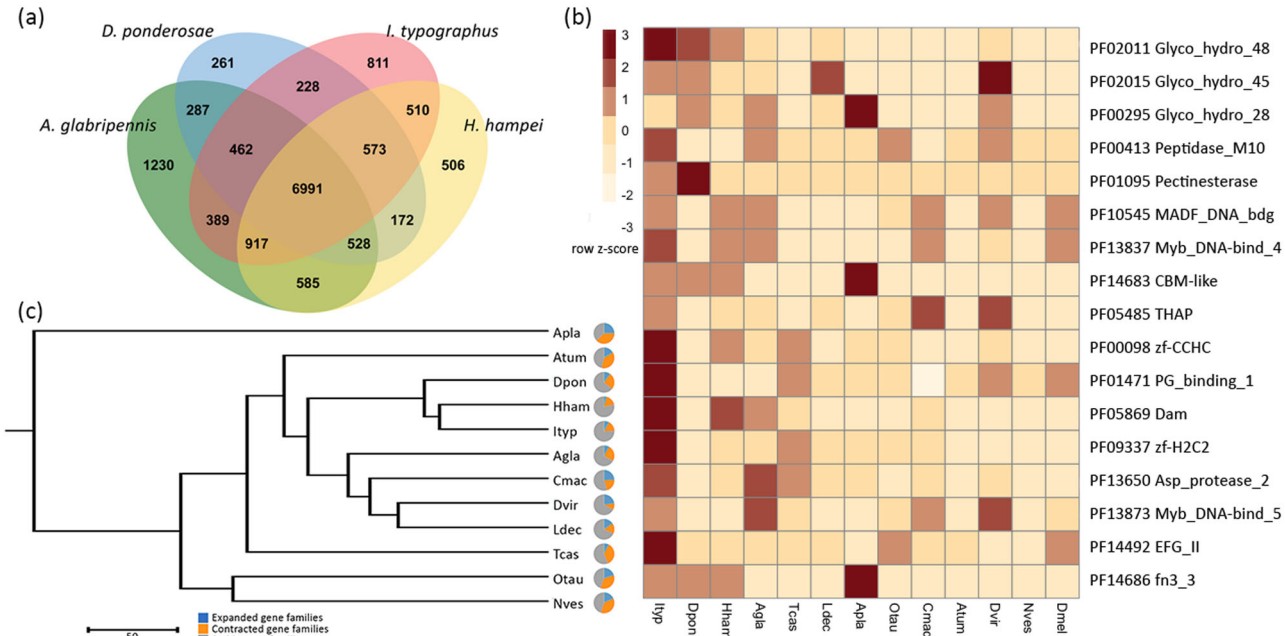

**Fig. 3 Gene ortholog analysis. a** Venn diagram of orthologous genes shared between *Ips typographus, Dendroctonus ponderosae, Hypothenemus hampei* (three bark beetles) and *Anoplophora glabripennis* (polyphagous cerambycid wood-borer). **b** Expanded protein families in *I. typographus* compared with other sequenced Coleoptera species and *Drosophila melanogaster*. Full names of protein families and corresponding PFam identifiers are given in Supplementary Table 7. Cell colours indicate the number of standard deviations from the mean of each domain count for all species. **c** Phylogenetic relationship of 11 Coleoptera species inferred from orthologous gene groups and including the number of expanded and contracted orthologous groups as identified by the CAFE analysis. Species abbreviations: Atum, *Aethina tumida*; Apla, *Agrilus planipennis*; Agla, *Anoplophora glabripennis*; Cmac, *Callosobruchus maculatus*; Dpon, *Dendroctonus ponderosae*; Dvir, *Diabrotica virgifera*; Hham, *Hypothenemus hampei*; Ityp, *Ips typographus*; Ldec, *Leptinotarsa decemlineata*; Nves, *Nicrophorus vespilloides*; Otau, *Onthophagus taurus*; Tcas, *Tribolium castaneum*; Dmel, *Drosophila melanogaster*. Scale bar represents 50 million years.

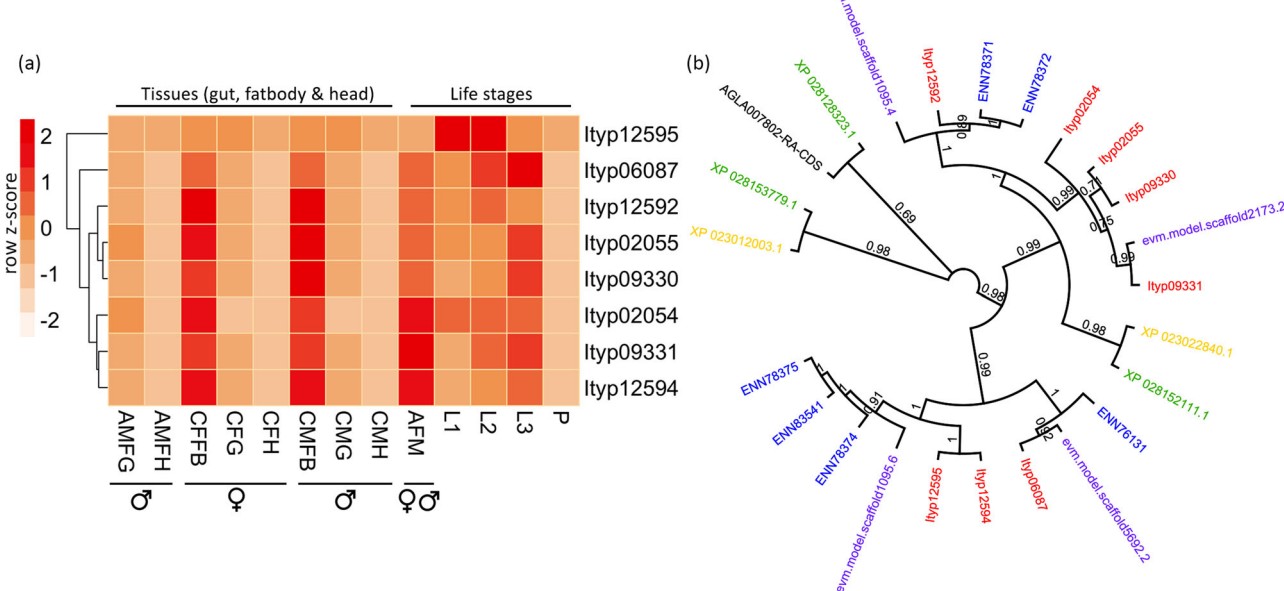

**Fig. 4 Expansion of GH48 domain containing proteins. a** Expression of the expanded GH48 domain (PF02011) containing proteins in 12 *Ips typographus* transcriptomes. AMFG, fed adult male gut; AMFH, fed adult male head; CFFB, callow female beetle fat body; CFG, callow female beetle gut; CFH, callow female beetle head; CMFB, callow male beetle fat body; CMG, callow male beetle gut; CMH, callow male beetle head; AFM, adult beetle (male & female); L1, larvae stage 1; L2, larvae stage 2; L3, larvae stage 3; P, Pupae. Cell colours indicate the number of standard deviations from the mean expression level. **b** Phylogenetic tree of GH48 domain-containing proteins in *I. typographus* (red) and five other Coleoptera species (blue text, *Dendroctonus ponderosae*; purple text, *Hypothenemus hampei;* black text, *Anoplophora glabripennis;* green text, *Diabrotica virgifera;* yellow text, *Leptinotarsa decemlineata*).

**Table 3 Comparison of protein families associated with detoxification and pesticide resistance between *Ips typographus* and 11 other Coleoptera species. The number of proteins containing the specific Pfam domain for each family is given.**

| Pfam domain | | Scolytini | | | Other plant feeders | | | | | | Other ecology | | |
|---|---|---|---|---|---|---|---|---|---|---|---|---|---|
| | | Ityp# | Dpon# | Hham* | Agla* | Tcas* | Ldec* | Apla* | Cmac* | Dvir* | Otau | Atum | Nves |
| Lig_chan | Ligand-gated ion channel | 33 | 17 | 31 | 56 | 45 | 40 | 45 | 40 | 55 | 56 | 35 | 54 |
| p450 | Cytochrome P450 | 84 | 93 | 64 | 120 | 133 | 114 | 109 | 132 | 238 | 182 | 124 | 156 |
| GST_N | Glutathione S-transferase, N-terminal domain | 31 | 24 | 16 | 24 | 35 | 29 | 20 | 16 | 36 | 43 | 44 | 24 |
| GST_C | Glutathione S-transferase, C-terminal domain | 27 | 22 | 11 | 18 | 32 | 22 | 20 | 11 | 33 | 29 | 32 | 21 |
| ABC_tran | ABC transporter | 75 | 74 | 72 | 77 | 89 | 129 | 106 | 140 | 166 | 115 | 83 | 82 |
| UDPGT | UDP-glucoronosyl and UDP-glucosyl transferase | 27 | 23 | 26 | 62 | 29 | 50 | 57 | 46 | 67 | 48 | 60 | 47 |
| COesterase | Carboxylesterase family | 59 | 63 | 49 | 105 | 56 | 118 | 80 | 106 | 168 | 88 | 59 | 62 |
| Inhibitor_I29 | Cathepsin propeptide inhibitor domain (I29) | 23 | 22 | 17 | 19 | 16 | 37 | 6 | 37 | 76 | 8 | 24 | 7 |
| | Total numbers | 359 | 338 | 286 | 481 | 435 | 539 | 443 | 528 | 839 | 569 | 461 | 453 |

Atum *Aethina tumida*, Apla *Agrilus planipennis*, Agla *Anoplophora glabripennis*, Cmac *Callosobruchus maculatus*, Dpon *Dendroctonus ponderosae*, Dvir *Diabrotica virgifera*, Hham *Hypothenemus hampei*, Ityp *Ips typographus*, Ldec *Leptinotarsa decemlineata*, Nves *Nicrophorus vespilloides*, Otau *Onthophagus taurus*, Tcas *Tribolium castaneum*. #Conifer feeders; *Angiosperm feeders; others, non-herbivores.

found in *I. typographus* is higher than in any of the other species included in this study, and clearly distinguishes *I. typographus* from *D. ponderosae* that has only a single gene containing this domain. Aspartyl protease is a digestive protease and has been implicated in proteolytic digestion in insects[38]. Both pectinesterase (PF01095) and polysaccharide lyase (PF14683) gene families were enriched in the three bark beetles. GH28 genes (PF00295) were also expanded in the bark beetles, but also in the wood-borer *A. glabripennis* and especially in the emerald ash borer, *A. planipennis* (Buprestidae). GH28 gene families are known to be involved in the degradation of pectin, a major component of primary plant cell walls. In several beetle species (*D. ponderosae*, *H. hampei*, *A. glabripennis*, and *L. decemlineata*) there is evidence suggesting a fungal origin of the GH28 genes[39], references therein. Expansions of genes involved in cell wall degradation may extend the capacity of these species to digest their recalcitrant host plants efficiently.

Searches of protein families within 12 species of Coleoptera revealed differences in the repertoire of genes important for detoxification and pesticide resistance. Interestingly, comparison between *I. typographus* and *D. ponderosae* revealed many similarities in gene family abundance (Table 3). Considering the well-known copious chemical defence by oleoresin of living conifer stems when invaded by bark beetles[40,41], one would hypothesise that bark beetles would have an increased number of genes involved in xenobiotic metabolism when compared to other agriculturally important coleopterans. In contrast, the two conifer-infesting bark beetles did not display an increased number and neither did the angiosperm-feeding *H. hampei* (Table 3). For the nine other coleopterans, the mean total Pfam number was $528 \pm 126$ (9), while the mean of the three bark beetles was $327 \pm 38$ (3), a difference of $-200$ (giving a strong Hedges $g$ standardized effect size value of $-1.6$ with a 95% C.I. $= -3.1$ to $-0.2$[42]). In particular, the number of cytochrome P450 genes in the *I. typographus* genome (86 genes) is lower than in all the other species included in our analysis, except for *H. hampei* (67 genes). However, the number of P450 genes in *D. ponderosae* is only marginally higher than in *I. typographus* (93 genes), and many of these genes were shown to form species-specific expansions mainly within the CYP6 and CYP9 subfamilies[26]. Hence, it is possible that Scolytinae specialists on conifers detoxify host defences (such as abundant monoterpenes and phenolics) using a smaller, but specialised, repertoire of P450 enzymes, as compared to generalist beetles feeding on diverse angiosperms. Conifers in the pine family (Pinaceae), in particular *Pinus* spp., are rich in preformed terpenes, and conifer specialists might therefore be expected to have numerous P450s to detoxify these. However, *I. typographus* seems to be less tolerant to terpenes than *Dendroctonus* spp.[43]. This may appear paradoxical considering its lifestyle, but *I. typographus* may have a strategy of quickly reducing the levels of tree defences by mass-attack and assistance from fungal symbionts[44]. Correspondingly, Norway spruce trees have less preformed defences and rely more on their capability for induced response for their resistance against bark beetle attacks[41].

## Conclusions

We assembled and annotated the genome of the Eurasian spruce bark beetle *I. typographus*, an important pest of spruce across the Palearctic. This highly contiguous, phased assembly comprises 236.8 Mb of sequence in 272 contigs with an $N_{50}$ of 6.65 Mb and contains 23,923 annotated genes. Our analyses of this assembly suggest gene family expansions of primarily plant cell wall degrading enzymes, of which aspartyl protease domain-containing genes stand out as particularly expanded. P450 enzyme-encoding genes involved in hormone synthesis and

catabolism of toxins are reduced in numbers. This whole-genome sequence from the genus *Ips* provides timely resources for population genetic studies and studies addressing important questions about the evolutionary biology and ecology of the true weevils. The genome is also expected to facilitate both fundamental and applied studies of functional genomics, as well as gene knockdown/knockout experiments. The essential odorant receptors are one example of a gene family that could be targeted to limit outbreaks, since interfering with the beetles' odour-guided host and mate finding behaviours likely will reduce their reproductive success[15,45]. Hence, this genomic resource may serve as a basis for improved and innovative management of an increasing threat to stressed trees in the Anthropocene.

## Methods

**Insect rearing and nucleic acid extraction**. To reduce heterozygosity in the beetles before PacBio long-read sequencing, we ran 10 generations of strict sibling mating. The first mating pairs were taken from a laboratory-reared population of *I. typographus* originating from Lardal, Norway. The continuous culture was kept at the Swedish University of Agricultural Sciences in Alnarp, Sweden on local, freshly cut Norway spruce, following the procedures described by Anderbrant and co-workers[46]. Sexes were separated with a pronotum hair density[47] and each mating pair had a $28 \times 12{-}14$ cm (length × diameter) spruce bolt providing food ad libitum[46]. For each new sibling mating generation, we used siblings from offspring from one or occasionally two bolts (strictly separated), ensuring strict sibling mating. To reduce the risk of breeding failure, the 10 successful sibling mating generations were undertaken by using at least 10 spruce bolts per generation, with little or no observed inbreeding depression effects on fecundity or body mass over time. High molecular weight DNA was extracted from around 100 adult males from this 10× inbred population to be used for high-throughput sequencing. Beetles were directly frozen in liquid nitrogen and pulverised using a pre-chilled mini-mortar and pestle. The resulting powder was transferred with a pre-chilled spatula to a microfuge tube containing a modified lysis buffer (20 mM EDTA, 100 mM NaCl, 1% Triton® X-100, 500 mM Guanidine-HCl, 10 mM Tris, pH 7.9) and ball bearings ("stainless steel beads", Qiagen). The sample was further homogenised using a Qiagen TissueLyzer by shaking for 1 min at 50 Hz. The optimal method for isolation of good quality high molecular weight DNA from *I. typographus* was using the Qiagen Genomic Tip 100/G kit following the manufacturers' instructions, except for a prolonged incubation time at 50 °C overnight with gentle agitation using the modified lysis buffer described above instead of the standard Qiagen G2 buffer, and with the addition of Proteinase K to 0.8 mg/ml. This was supplemented with the addition of DNase-free RNase A (20 μg/ml) treatment and incubation for 30 min at 37 °C, followed by centrifugation for 20 min at $12{,}000 \times g$ to pellet insoluble debris. The clarified lysate was then transferred to the QBT buffer-equilibrated Genomic Tip to proceed with the standard protocol.

**Genome sequencing, assembly, and evaluation**. DNA samples were transported to the sequencing facility at the Uppsala Genome Center, Science for Life Laboratory, Uppsala University, Sweden for library preparation and sequencing. A pooled DNA sample from the 10th generation inbred population was used to generate SMRT cell libraries that were sequenced on the PacBio RS II platform (Pacific Biosciences). DNA from a single non-inbred male beetle from the continuous laboratory culture described above was used to generate a 10X chromium library that was sequenced using the HiSeq X system (Illumina) to produce short-read data for a genome survey (364 million 150 bp reads, ~230-fold genome coverage). Primary assembly of the PacBio long reads was performed using FALCON-kit v1.3.0 software and haplotype phasing was achieved using FALCON-unzip v1.2.0 with default settings[48]. Contigs from the final assembly were aligned against each other using MUMmer[49] and redundant contigs removed. The completeness of the genome assembly was assessed using metrics derived from read alignment (Supplementary Fig. 5) and searches of single-copy orthologs. The Benchmarking Universal Single-Copy Orthologs (BUSCO v3.0.2)[50] tool was used to search the genome assembly against the insecta_odb9 database of 1,658 genes. In order to obtain an understanding of the chromosomal structure of long contigs, telomeric motifs were identified using the script FindTelomeres.py (https://github.com/JanaSperschneider/FindTelomeres). Genome size estimation was performed using data from the short-read genome survey and *k*-mer counting with Jellyfish v2.3.0[51] using GenomeScope 2.0[52].

**Transcriptome assembly and quantification of gene expression**. Transcriptome libraries from different *I. typographus* life stages and tissue types (larval stages 1-3, pupae, adult beetle, male head, female head, callow male gut and fat body, callow female gut and fat body) were prepared at the Czech University of Life Sciences, Prague (Supplementary Table 8). All beetle samples were taken directly from logs obtained from Rouchovany, Czech Republic. Beetles were starved overnight before dissection or preservation for downstream processing. Specifically, whole beetles in

different life stages (larvae, pupae, adults) were snap-frozen in liquid $N_2$, and specific tissues (gut, head, fat body) were dissected under sterile conditions and kept in RNA*later*™ until RNA extraction. Total RNA from whole larvae, pupae and adult *I. typographus* and the different tissues (pooled from several adults) was purified using the PureLink™ RNA Kit from Ambion (Invitrogen) following the manufacturer's protocol. The integrity of the purified total RNA was checked using agarose gel electrophoresis and the 2100 Bioanalyzer system (Agilent Technologies, Inc). Total RNA samples with an integrity number (RIN) > 7 were selected for sequencing. Total RNA was quantified using a Qubit 2.0 Fluorometer (Thermo Fisher Scientific) with an RNA HS Assay Kit (Invitrogen) before sending for sequencing at Novogene, China. After the initial quality control, mRNA from eukaryotic organisms was enriched using oligo(dT) beads and cDNA libraries were prepared using the NEB Next® Ultra™ RNA Library Prep Kit and sequenced on the Illumina platform to generate a minimum of 30 million 150 bp paired-end reads per sample. Five biological replicates for each sample type were sequenced. Raw reads were processed using Trimmomatic v0.36[53] and assembled using Trinity v2.8.2[54] with the default parameters except for --normalize_max_read_cov 100 --min_kmer_cov 3 --min_glue 3 --KMER_SIZE 23. Expression levels were measured by aligning quality-processed RNA-Seq reads to the genome assembly using HiSat2 v2.1.0. Aligned reads were sorted with SAMtools v1.5[55] and counts reported as transcripts per one million mapped reads (TPM) using StringTie v1.3.3. TPM values visualised in heatmaps were transformed to $\log_2 (\text{TPM} + 1)$ and normalised across tissues using the scale function in R[56].

**Genome annotation**. A custom repeat library was produced from the genome assembly using RepeatModeler v1.0.11 (http://www.repeatmasker.org/RepeatModeler/). This custom library, together with the Repbase library, was used with RepeatMasker v4.0.8 (http://www.repeatmasker.org/) to soft mask the assembly prior to annotation. RNA-Seq data from multiple beetle life stages and tissues (Supplementary Table 8) were aligned to the genome assembly using HiSat2 v2.1.0[57] and annotated using StringTie v1.3.3[58]. The alignment features were passed to BRAKER2[59] as hints for training an AUGUSTUS model using GeneMark-ET. The transcriptome assembly from the RNA-Seq data was used together with the published protein sequences from the genomes of three species of Coleoptera, namely *Dendroctonus ponderosae* (Scolytinae), *Anoplophora glabripennis* Motschulsky (Cerambycidae), *Tribolium castaneum* Herbst (Tenebrionidae), and also the protein sequences from a further eight high-quality genomes, including *Drosophila melanogaster* Meigen (Diptera) and *Homo sapiens* Linné, as evidence for homology-based gene prediction using MAKER3[60], as outlined in Supplementary Figure 6. Because transcriptome data is prone to misassembly and chimeras it was only used for evidence and not for prediction. Protein sequences identified from the genome-level BUSCO searches were passed to the MAKER3 pipeline for training SNAP[61] along with the BRAKER2-trained AUGUSTUS model for ab initio predictions, using the HiSat2 RNA-Seq alignments as transcript-based evidence. Gene model predictions were retained if they were greater than 33 amino acids in length and were supported by at least one form of supporting evidence of either (i) a BLASTp match ($E$-value $< 10^{-10}$) to the non-redundant protein GenBank database, (ii) coverage by RNA-Seq data alignment, or (iii) containing a hit to the Pfam database. Several iterations of MAKER3 were performed to improve upon the prediction in subsequent runs. BUSCO analysis was subsequently performed on the predicted protein sequences using the same parameters as listed in Section 2.2, except this time running in protein mode.

**Comparative genomics and phylogenomics**. In order to identify expansions of gene families within the *I. typographus* genome, we searched the protein-coding sequences for Pfam[62] domains to assign gene function. We used HMMER v3.1[63] (hmmscan) to search the Pfam A database (release 32.0) for 13,841 different domains of 20 different species of Metazoa. Counts of each domain were collated for each species and domains that occurred multiple times in a protein sequence were counted only once. A Fisher's exact test was then conducted iteratively using R, comparing the number of counts in Pfam families found in an individual genome. Counts were normalised by the total gene count for each species against the background, which we defined as the average of the counts in the remaining species. Multiple testing corrections were done using the False Discovery Rate (FDR) method in R for all calculated $p$-values. A Pfam domain was considered expanded in *I. typographus* if it showed a corrected $p$-value $< 0.05$. This approach has been used in several recent papers[64–66].

Protein sequences from *I. typographus*, *D. ponderosae*, *T. castaneum*, *H. hampei* and *A. glabripennis* were compared for orthology using all-against-all alignments ($E$-value of $10^{-5}$) and clustered using OrthoVenn2[67] with an inflation value of 1.5. The unique set of 811 *Ips*-specific gene clusters identified from this comparison were assigned gene ontology terms using the Uniprot database and gene ontology term enrichment was computed using a hypergeometric distribution in OrthoVenn2. Phylogenetic inference from orthologous genes was undertaken by comparing the *I. typographus* gene set with the protein-coding sequences from 11 other species of Coleoptera (Supplementary Table 6) using OrthoFinder v2.4.0[68]. Using this package, single copy orthologous gene groups were used to build an ultrametric maximum likelihood-rooted species-tree from multiple sequence alignments using the inbuilt MAFFT and FastTree options. The resulting tree was then visualised using FigTree v1.4.4 (https://github.com/rambaut/figtree/releases).

We used CAFE v4.2.1[69] using the default *P*-value thresholds to examine the expansion and contraction of the orthologous gene groups identified with OrthoFinder.

**Statistics and reproducibility**. Presence of genes associated with detoxification and pesticide resistance in sequenced species of Coleoptera were identified based on Pfam domains contained within their respective protein sequences. For comparison of the mean Pfam numbers of the three scolytines and nine other Coleoptera we calculated Hedges *g* standardized effect size, which compares a pair of means by scaling their difference by division of their pooled standard deviations, using conventional methods[42,70,71]. Other statistical tests used in this study are described in the relevant methods section. All analyses presented in this study can be reproduced from the data deposited in publicly available databases as described in the data availability section.

**Reporting summary**. Further information on research design is available in the Nature Research Reporting Summary linked to this article.

## Data availability

This Whole Genome Shotgun project has been deposited at DDBJ/ENA/GenBank under the accession JADDUH000000000. The version described in this paper is version JADDUH010000000. All sequence data relating to this study are available under the BioProject accession numbers PRJNA671615 and PRJNA679450. Accompanying data, including source data used to generated figures, is permanently available from Figshare https://doi.org/10.6084/m9.figshare.14503065. All other data are available from the corresponding author on reasonable request.

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

## Acknowledgements
We would like to acknowledge the support of the National Genomics Infrastructure (NGI)/Uppsala Genome Center and UPPMAX for providing assistance in massive parallel sequencing and computational infrastructure. Work performed at NGI / Uppsala Genome Center has been funded by RFI/VR and Science for Life Laboratory, Sweden. We thank several colleagues for discussions and help on genome sizes, chromosomes, and long contigs data: J. Spencer Johnston (Texas A&M University, College Station), Anthony I. Cognato (Michigan State University, East Lansing), and Krystyna Nadachowska-Brzyska & Piotr Zieliński (Jagiellonian University, Krakow). Dr Jan Bílý (Czech University of Life Sciences, Prague) is acknowledged for technical support during RNA samples preparation for tissue transcriptome study. Infrastructural support and salary for A.R., E.G-W. and F.S. were obtained from project EXTEMIT-K CZ.02.1.01/0.0/0.0/15_003/0000433 financed by OP RDE at Czech University of Life Sciences, Prague. M.N.A. was funded by the Swedish Research Council FORMAS (grants #217-2014-689 and #2018-01444). C.L. acknowledges support from the Swedish Research Council VR (grant #2017-03804). P.K. was funded by the Research Council of Norway (grant #249958/F20).

## Author contributions
D.P. performed the genome annotation, all bioinformatic analyses and wrote the manuscript, with M.N.A contributing to the initial draft. M.N.A. and F.S. conceived, initiated, and designed the study with E.G-W. and P.K., F.S. performed the sib-mating process with A.C., H.V. performed DNA extraction, C.L. provided resources for the bioinformatics, and A.R. and E.G-W. undertook tissue transcriptome sequencing for genome annotation. F.S. performed the effect size analysis. All authors commented on, improved and approved the final manuscript.

## Funding

## Competing interests
The authors declare no competing interests.
