## [Peer Review File · Communications Biology]

Reviewers' comments:

Reviewer #1 (Remarks to the Author):

The manuscript from Powell et al. was a clear description of a genome assembly of *Ips typographus* along with some nice comparative genomic analyses looking at gene families. I found the methods to be well described and I have no major concerns with the paper. I agree with the authors that more high quality beetle genomes are needed, and their assembly is a nice contribution on that front. I have a few minor issues that the authors may want to consider before publishing. In no particular order.

- 1) Was there any assessment of molecular genetic variation in the final sequenced line? It can be difficult to remove all variation and some species can still have large tracts of variation in the genome even after long periods of inbreeding. This can be due to structural variation (inversions) and linked lethals which exist in pretty much every inbred line I've looked at. These patterns of variation across the genome (low in some areas, high in others) can lead to breaks in the assembly. Do the authors think there are any inversions or regions of high variation? It would be very interesting to see the PacBio data mapped back to the assembly (or Illumina data from an inbred individual) and variants called and their density plotted along the contigs.
- 2) Has there been any cytological work to determine how many chromosomes this species has and what the chromosomes look like? Are they metacentric, telocentric? Although not necessary a chromosome squash would be very helpful here and could provide some hints as to how close you are getting to chromosome level. Additionally, I'm guessing HiC is coming for this species to bring it up to chromosome level, in which case the squashes will be necessary.
- 3) 220-222 I would consider rewording just to say that the largest contigs include telomere sequence. You might already be assembling both arms of acrocentric chromosomes (small contigs have telomere sequence too), in which case the small contigs with telomere are in many ways just as impressive and good as the long arm of the chromosome.
- 4) The total gene model number is higher than other coleopteran genomes (*A. glabripennis* is close I suppose). Do you have an explanation as to why? The difference between *D. ponderosae* and *I. typographus* is significant.
- 5) Table 1 suggests that you have a fair number of 'complete and duplicated' genes from your BUSCO analysis. Admittedly, these could be real, but I would strongly recommend an Illumina coverage analysis of the genome assembly. Using Illumina data generated from a female that is then mapped to your draft, all contigs in the assembly should have pretty much the same coverage when averaging depth in windows across the genome. Upward deviations indicate collapsed parts of the assembly (often seen in repetitive regions) or significant dips could be things that should have been collapsed during assembly but were not. I would follow up and find those BUSCOs that are duplicated and see if they colocalize on contigs or regions of the genome where coverage dips down relative to neighboring regions or is less than other contigs throughout.
- 6) Although not necessary, I would also consider plotting repeat density across your largest contigs to see if density increases on the end opposite the telomere sequence (See <https://genomebiology.biomedcentral.com/articles/10.1186/gb-2008-9-3-r61>). Would be an indication you are entering into pericentromeric sequence and the contig is 'whole arm'.
- 7) Line 107. Were these all one sex or both sexes mixed? If mixed sex or all males you will have lower coverage on the X chromosome (see 1 and 5)

Reviewer #2 (Remarks to the Author):

Report on manuscript "A highly contiguous genome assembly of a major forest pest, the Eurasian spruce bark beetle *Ips typographus*" by D. Powell and collaborators.

The manuscript presents new genomic resources for the pest *Ips typographus*, a spruce associated bark beetle that can cause significant damage to European forests. The authors have obtained a high-quality genome assembly from an inbred line of the beetle, and have annotated this resource thanks to a gene prediction approach and the use of original RNA-seq data. Enrichment analyses

were performed. 22 gene families were expanded in this species, and a detailed discussion is provided.

The manuscript is clear and well organised, the analyses done are well presented. I only have minor comments which are detailed below.

The Introduction section is clear. As details are given about the chemical cues of aggregation (lines 59-63), I was expecting that some of the identified genes would be related to pheromone production or chemical signalling, which is not the case. Conversely, most of the discussion is about genes involved in wood degradation, which is expected in a bark beetle species but is not mentioned in introduction. The authors might slightly modify the introduction to make it fit better with manuscript contents

Material and Methods

line 98 : correct formatting of Anderbrant's reference (29)

line 106 : were the beetles directly frozen in liquid nitrogen or were they first stored in any buffer ?

line 108 : the recipe of the lysis buffer should be given here, at first mention, rather than line 112.

line 108 : do you really mean ball bearing ? Give a reference

lines 133-134 : was the Illumina sequencing performed using the same DNA than the PacBio ?

Illumina sequencing should be described at the beginning of the 2.2 section, with details concerning library preparation and sequencing center.

lines 140-142: you mention these RNAseq data before the paragraph describing the transcriptome sequencing and assembly. You should move the details concerning stage and tissue sampling and RNAseq experiment before presenting how you used these reads for genome annotation.

lines 178-179 : I think that the phylogenetic tree does not bring much to the manuscript. It is actually not discussed. I suggest deleting it.

lines 185-190 : it could be useful to provide more details about the sampling used for RNAseq experiment. Were the larvae and adults taken from the same rearing ? Were they also sampled after 10 inbred generations ? Did the larvae starve for some time before extraction ? Were the adults allowed to fly or sampled directly in the logs ?

Results and Discussion

Line 233 : how many chromosomes are expected in a bark beetle species ?

lines 242-243 : The Material and Methods section mentions a Busco analysis to estimate genome completeness (corresponding to the results given lines 225-227) but the Busco analysis done on the predicted gene set is not clearly described in the M&M.

line 266 : why and how did you select 17 of the expanded gene families out of the 22 ? It is possible to show the 22 in Fig 3b. Provide the complete names of the gene families

lines 271-272: as mentioned earlier, the species phylogenetic tree does not bring relevant information and is not discussed. It could be deleted.

Reviewer #3 (Remarks to the Author):

Daniel Powell et al. in "A Highly contiguous genome assembly of a major forest pest, the Eurasian spruce bark beetle *Ips typographus*" sequenced the genome using PacBio where they report a phased contiguous haploid assembly of 236 Mb of sequence and N50 of 6.65 Mb. They annotated the genome where they report 23,923 genes. They suggest with parsimonious analysis that gene family such as plant cell wall degrading enzymes; aspartyl protease domain containing genes are expanded. The author finds reduction in genes responsible in catabolism of toxins. They report the first whole-genome sequence from the genus *Ips* and allude to the importance of this genome to tackle evolutionary and ecology questions in this pest as well as design genetic manipulation experiment for its eradication.

The manuscript read well, but I have few minor comments on the manuscript that need to be addressed:

1- line 133. The author used a kmer approach to estimate genome size, which is GenomeScope

2.0 relying on jellyfish analysis. First, is there a flowcytometry measure for *Ips typographus* and how your kmer estimate will compare to that? Second, using their kmer approach, if it is possible to mention which kmer they use for this analysis. It would be nice if they run multiple kmers and show some GenomeScope figure results, with heterozygosity estimates. Third, in case there is no flowcytometry measure available, it would be great to have the author run another tool such as KmerFreq_AR in SOAPdenovo2 at similar kmers to see if they get the same results as GenomeScope.

2- Line 223. The author report 28.2 % of the *Ips typographus* made of repeats. They used Repeatmasker, which will generate a report for the different classes of repeats (Long terminal repeats (LTR), DNA etc..). Since it is about the genome, and comparative genomics. It is better to report the different classes of TEs from RepeatMasker and do the comparison with other published genomes reported in Supplementary Table 2. Is there any correlation between genome size and TEs?

3- The author discuss in Line 283 and 298, which GH48 is part of the *Ips typographus* genome, but also they talk about horizontal gene transfer from bacteria or fungi. In addition, they cite that gut microbiome of *Ips typographus* is the place to help the species cope with its enzymatic digestion during colonization. I would like to ask if the author did any analysis to rule out its horizontal transfer origin from fungi? They have mentioned phylogentic analysis of GH48, which has similarity to other Scolytinase species *D. ponderosae* and *H.hampei*. Did they search similarly to Fungi database?

4- The author did comparative genomics and Phylogenomics line 161....171 and report expansion and contraction of gene families using a parsimonious approach and Fisher exact test. Usually, statistical significance of the family size changes should be inferred across a species phylogeny using a stochastic birth-death process based model such as the one implemented in CAFÉ from De Bie et al. 2006. I was wondering if the authors can run their analysis using this tool, would they expect the same results for the expanded and contracted gene families as they have reported?

The authors have to address these points, and then it will be considered for publication.

Reviewer #4 (Remarks to the Author):

This paper reported the genome of the Eurasian spruce bark beetle (*Ips typographus*), which is one tree-killing species and causes economic and ecological costs. Decoding the genome of this beetle is helpful to develop novel pest control strategies to mitigate its outbreaks in forest. However, to get results with higher quality and thus provide deeper insights into this species, some analysis strategies should be changed and more works should be done. Questions and problems are listed below.

1. The results of completeness evaluation of this draft genome through BUSCO tool were not consistent in this manuscript. From the line 225 to 227, the author stated that "1,649 (99.5%) of the 1,658 genes in the insecta_odb9 database could be identified as present either partial or complete", while from the line 241 to 244, the author stated that "searches of the annotated predicted protein sequences using BUSCO from the insecta_odb9 libraries identified 1,584 (95.5%) of the 1,658 protein sequences in the Insect dataset". What induced this difference, and what is the correct one?

2. From the line 250 to 252: "an overall average of 93.6% mapped reads indicates that the majority of transcripts captured in our RNA-seq data can be found and are intact in the *I. typographus* gene set". Only mapping rate of RNA-seq data can't indicate the intact structure of gene models, thus this statement is not very reasonable.

3. From the line 262 to 264, the author performed a gene ontology enrichment analysis for the unique set of 811 gene clusters in *I. typographus* and found some enriched functions. First, the method adopted in this analysis should be clarified. Second, these enriched functions seem very general and important for any animals. Thus, more details should be listed and discussed to reveal

the uniqueness of these enriched functions; otherwise it is tended to suspect there are some problems in the identification of unique set of 811 gene clusters in *I. typographus* due to technique errors.

4. From the line 264 to 266, the author identified 22 gene families which were significantly expanded in *I. typographus* when compared with the 11 other coleopteran genomes and a selection of other model invertebrates. Expanded gene families in *I. typographus* were identified taking the average member counts of these gene families in remaining species as the background through Fisher's exact test. I challenge that the average member counts of gene families in the remaining species shouldn't be considered as the background, because the average statistical value is misleading in this context. Expansion or contraction of gene families in a species is a relative status compare to the most recent common ancestor (MRCA), whereas the average member counts of gene families in remaining species can't represent their ancestral status. The CAFE software package should be used to identify expanded gene families.

5. From the line 267 to 269, the author stated that "the limited number of significantly expanded gene families discovered in *I. typographus* in this analysis reflects the comprehensive array of beetle species included in the dataset". This logic is hard to follow, because the number of expanded or contracted gene families is related to the evolutionary pattern but not the sampling strategy.

6. From the line 275 to 277, the author identified significantly expanded gene families associated with plant cell wall degrading enzymes in *I. typographus*. Similar to the problem I mentioned above, expanded gene families should be identified by the CAFE software package.

7. From the line 286 to 288, the author described the surprised expressed pattern of glycosyl hydrolase (GH) enzymes in *I. typographus* (which is predominant in the fat body), but it failed to provide more detailed results and citations to support or explain this phenomenon.

8. From the line 297 to 300, the author found "three separate tandem duplication events appear to have occurred in *I. typographus* since the establishment of GH48 genes in the Curculionoidea, seen as three pairs of genes occurring adjacent to each other in the genome (sequentially numbered gene models are found adjacently in the genome)". First, the physical distance and sequence similarity between these three pairs of genes should be displayed with a table or a figure. Genes generated through tandem duplication events are close to each other in the physical distance and similar to each other in the gene sequence. Only sequential number of these gene models is not enough to indicate tandem duplication events. Second, a synteny plotting among species in the Curculionoidea should be provided to reveal whether the tandem duplication events of this gene happened in other related species. Otherwise, the conclusion of "three separate tandem duplication events appear to have occurred in *I. typographus* since the establishment of GH48 genes in the Curculionoidea" can't be made.

9. The citation information is not complete in the line 316.

10. From the line 323 to 325, the author stated "these two conifer-infesting bark beetles did not display an enrichment for gene families involved in xenobiotic metabolism when compared to other agriculturally important Coleopterans". This conclusion can't be made without any evidences got from statistical test. Although the member counts of these gene families in *I. typographus* are smaller than other species, the background of other gene families, the quality of genome assemble as well as gene annotation and whether the difference is significant should be considered.

11. From the line 329 to 332, the author stated "it is possible that Scolytinae specialists on conifers detoxify host defences (such as abundant conifer monoterpenes and phenolics) using a smaller, but specialized, repertoire of P450 enzymes, as compared to generalist beetles feeding on diverse angiosperms". And from the line 335 to 337, the author suggested "this may appear paradoxical considering its lifestyle, but *I. typographus* may have a strategy of quickly reducing the levels of tree defences by mass-attack and assistance from fungal symbionts". Both of these two statements are built on the conclusion that there is no enrichment in *I. typographus* for gene families involved in xenobiotic metabolism. As is mentioned above, this conclusion is not solid.

Further, to make a whole research, some experiments associated with functional genomics should be done, such as cloning some key genes coding P450 enzymes and testing their activity. In addition, 16S rDNA in the gut of *I. typographus* should be sequenced to reveal the pattern of its fungal symbionts.

12. From the line 352 to 355, the author concluded that "the essential odorant receptors are one example of gene family that could be targeted to limit outbreaks, since interfering with the beetles' odour-guided host and mate finding behaviours likely will reduce their reproductive success". However, there is no any related analysis been done in this manuscript, and it should be added to make deeper insights into this species *I. typographus*.

13. From the line 143 to 149, the author adopted the transcriptome de novo assembled from the RNA-Seq data of *I. typographus* through Trinity as evidence for homology-based gene prediction. In this way, assembled errors might be taken into the model used for gene prediction. Thus, it is more appropriate to use RNA-Seq data only as transcript-based evidence.

14. The line 178: "orthologous gene groups were used to build a rooted species tree". What kind of orthologous gene groups were adopted to build the rooted species tree? In fact, single-copy orthologous gene groups should be used.

15. From the line 185 to 187, the author prepared transcriptome libraries for callow male gut and callow female gut of *I. typographus*. It should be clarified that how to distinguish the transcriptomes of *I. typographus* and its gut symbiont.

16. The line 198: the version of Trimmomatic used in this analysis should be clarified.

Referee Comments	Author's response
Reviewer #1:	
The manuscript from Powell et al. was a clear description of a genome assembly of Ips typographus along with some nice comparative genomic analyses looking at gene families. I found the methods to be well described and I have no major concerns with the paper. I agree with the authors that more high quality beetle genomes are needed, and their assembly is a nice contribution on that front. I have a few minor issues that the authors may want to consider before publishing. In no particular order.	Thank you for the positive assessment!
1) Was there any assessment of molecular genetic variation in the final sequenced line? It can be difficult to remove all variation and some species can still have large tracts of variation in the genome even after long periods of inbreeding. This can be due to structural variation (inversions) and linked lethals which exist in pretty much every inbred line I've looked at. These patterns of variation across the genome (low in some areas, high in others) can lead to breaks in the assembly. Do the authors think there are any inversions or regions of high variation? It would be very interesting to see the PacBio data mapped back to the assembly (or Illumina data from an inbred individual) and variants called and their density plotted along the contigs.	The technology used at the time of this study was constrained by DNA quantity. Thus, we did not assess the level of genetic variation among individuals as the amount of DNA obtainable from a single beetle was insufficient. We appreciate that some small level of variation may have led to a decrease in the contiguity of our assembly, though we believe the assembly is of a level of quality commensurate with other recently reported insect species. It should be noted that the inbreeding was an absolute sib-mating regime for 10 generations.
2) Has there been any cytological work to determine how many chromosomes this species has and what the chromosomes look like? Are they metacentric, telocentric? Although not necessary a chromosome squash would be very helpful here and could provide some hints as to how close you are getting to chromosome level. Additionally, I'm guessing HiC is coming for this species to bring it up to chromosome level, in which case the squashes will be necessary.	The karyotype for Ips typographus is 14 + Xyp. We have included this in line 230-231 of the manuscript. We do not know whether chromosomes are meta- or telocentric. Hi-C scaffolding is planned to be undertaken in the future. However, it is not possible to be included as part of the present study. We appreciate that this would greatly improve our assembly, but this is not yet available to us and will be completed as a separate future study.
3) 220-222 I would consider rewording just to say that the largest contigs include telomere sequence. You might already be assembling both arms of acrocentric chromosomes (small contigs have telomere sequence too), in which case the small contigs with telomere are in many ways just as impressive and good as the long arm of the chromosome.	This sentence (Lines 239-240) has been edited to include only this statement as per the reviewer's suggestion.

4) The total gene model number is higher than other coleopteran genomes (A glabripennis is close I suppose). Do you have an explanation as to why? The difference between D. ponderosae and I. typographus is significant.	We accept there is a significant difference in the number of gene models between I. typographus and D. ponderosae. However, the number of gene models in I. typographus is comparable to the more closely related H. hampei (19,222 predicted genes). There is quite some variation throughout the coleopteran genomes, C. maculatus has 31,345 predicted genes and D. virgifera has 28,061 predicted genes further to A. glabripennis as mentioned (22,434 predicted genes), A. planipennis has 22,159 predicted genes and O. taurus has 21,668 predicted genes. We believe our estimates are in line with other similar species. In fact, D. ponderosae has the lowest number of predicted genes for any coleopteran genome assembly so far (with over 4,000 less genes than the next species).
5) Table 1 suggests that you have a fair number of ‘complete and duplicated’ genes from your BUSCO analysis. Admittedly, these could be real, but I would strongly recommend an Illumina coverage analysis of the genome assembly. Using Illumina data generated from a female that is then mapped to your draft, all contigs in the assembly should have pretty much the same coverage when averaging depth in windows across the genome. Upward deviations indicate collapsed parts of the assembly (often seen in repetitive regions) or significant dips could be things that should have been collapsed during assembly but were not. I would follow up and find those BUSCOs that are duplicated and see if they colocalize on contigs or regions of the genome where coverage dips down relative to neighboring regions or is less than other contigs throughout.	A small degree of duplication is quite common for this grade of assembly and that most reports tolerate this small level of error as it does not detract from downstream analysis. We have similar levels of duplicated BUSCOs as the D. ponderosae genome assembly and indeed much less than other coleopteran genomes published to date. While improvements could be made to the assembly to identify and remove artificially duplicated regions, our genome assembly is of a standard better than most for non-model insects using the same technology. We have, however, added a coverage plot of the 30 largest contigs to the supplementary materials (Supplementary Figure 5). It shows mostly even coverage across these regions apart from the contigs from the putative sex chromosome.
6) Although not necessary, I would also consider plotting repeat density across your largest contigs to see if density increases on the end opposite the telomere sequence (See https://genomebiology.biomedcentral.com/article/10.1186/gb-2008-9-3-r61). Would be an indication you are entering into pericentromeric sequence and the contig is ‘whole arm’.	This would be an interesting observation, though for this study, we also feel it is not necessary to support our modest claims. However, we take it as piece of good advice for our further study that will also include the addition of Hi-C data.
7) Line 107. Were these all one sex or both sexes mixed? If mixed sex or all males you will have lower coverage on the X chromosome (see 1 and 5)	All individuals in the sample were males (adults). Thanks for spotting that we had missed to include this information. We have added this information on Line 100.

Reviewer #2:	
1) The manuscript presents new genomic resources for the pest Ips typographus , a spruce associated bark beetle that can cause significant damage to European forests. The authors have obtained a high-quality genome assembly from an inbred line of the beetle, and have annotated this resource thanks to a gene prediction approach and the use of original RNA-seq data. Enrichment analyses were performed. 22 gene families were expanded in this species, and a detailed discussion is provided. The manuscript is clear and well organised, the analyses done are well presented. I only have minor comments which are detailed below.	Thank you for the positive assessment and valuable suggestions for improvements!
2) The Introduction section is clear. As details are given about the chemical cues of aggregation (lines 59-63), I was expecting that some of the identified genes would be related to pheromone production or chemical signalling, which is not the case. Conversely, most of the discussion is about genes involved in wood degradation, which is expected in a bark beetle species but is not mentioned in introduction. The authors might slightly modify the introduction to make it fit better with manuscript contents	We agree and have therefore shortened the paragraph about chemosensation and cut down on the details about pheromone production. Our intention with this paragraph was mainly to highlight that chemosensation and pheromone communication are central to the success of this species, and therefore genetic resources in terms of antennal transcriptomes have been generated. However, since genetic resources are essentially limited to these antennal transcriptomes, a genome assembly would be highly valuable. We have clarified this in the revised version (Lines 58-63).
3) Material and Methods line 98 : correct formatting of Anderbrant's reference (29)	This reference formatting error has now been corrected (Line 93).
4) line 106 : were the beetles directly frozen in liquid nitrogen or were they first stored in any buffer ?	The beetles were directly frozen, i.e., not stored in any buffer. We have clarified this on Line 101.
5) line 108 : the recipe of the lysis buffer should be given here, at first mention, rather than line 112.	This information has now been moved as per the recommendation of the Reviewer (line 103-104).
6) line 108 : do you really mean ball bearing ? Give a reference	Yes, ball bearings, also known as stainless steel beads, are used as part of the TissueLyzer system for mechanical cell disruption. These are specified in the manufacturer's instructions and are sold as a standard product. To clarify in the manuscript, we have added "stainless steel beads" and Qiagen (for reference) on Line 105.
7) lines 133-134 : was the Illumina sequencing performed using the same DNA than the PacBio ? Illumina sequencing should be described at the	Details of the samples used for sequencing on both the PacBio and Illumina platforms have now been added to the beginning of section

beginning of the 2.2 section, with details concerning library preparation and sequencing center.	2.2, lines 120-124.
8) lines 140-142: you mention these RNAseq data before the paragraph describing the transcriptome sequencing and assembly. You should move the details concerning stage and tissue sampling and RNAseq experiment before presenting how you used these reads for genome annotation.	Section 2.5 concerning the RNA-Seq data has now been moved to become section 2.3, prior to the section concerning genome annotation, as suggested (lines 136-162).
9) lines 178-179 : I think that the phylogenetic tree does not bring much to the manuscript. It is actually not discussed. I suggest deleting it.	Figure 3c has now been updated to also include the outcome of the CAFE analysis. We believe it also helps to inform the reader, who may not have a deep understanding of coleopteran systematics, of the degree of relatedness between the species included in this study.
10) lines 185-190 : it could be useful to provide more details about the sampling used for RNAseq experiment. Were the larvae and adults taken from the same rearing ? Were they also sampled after 10 inbred generations ? Did the larvae starve for some time before extraction ? Were the adults allowed to fly or sampled directly in the logs ?	These details have now been added to Lines 137-147. These beetles were not from the same inbred line as used for the PacBio sequencing. Specimens were taken from logs, starved overnight and then shock frozen in liquid N2. In case of specific tissue dissections, RNA later was used to preserve the nucleic acid until extraction.
11) Results and Discussion Line 233 : how many chromosomes are expected in a bark beetle species ?	The karyotype for Ips typographus is 14 + X _Y _p . We have included this in Lines 230-231 of the manuscript.
12) lines 242-243 : The Material and Methods section mentions a Busco analysis to estimate genome completeness (corresponding to the results given lines 225-227) but the Busco analysis done on the predicted gene set is not clearly described in the M&M.	Details describing the BUSCO protein-level analysis has been included at the end of section 2.4 (Lines 186-188).
13) line 266 : why and how did you select 17 of the expanded gene families out of the 22 ? It is possible to show the 22 in Fig 3b. Provide the complete names of the gene families	The other five gene families were domain model variations contained within the reported domains that we believe simply did not add value to the figure panel. The complete names of the gene families take up far too much space to present alongside the figure (up to 56 characters in a name). These details have now been included in the supplementary data (see Supplementary Table 7).
14) lines 271-272: as mentioned earlier, the species phylogenetic tree does not bring relevant information and is not discussed. It could be deleted.	This has now been revised to include outcome from CAFE analysis and we believe it adds value for readers not familiar with beetle taxonomy.
Reviewer #3:	
Daniel Powell et al. in "A Highly contiguous genome assembly of a major forest pest, the	Thank you for the positive assessment and valuable comments for improvements!

Eurasian spruce bark beetle Ips typographus” sequenced the genome using PacBio where they report a phased contiguous haploid assembly of 236 Mb of sequence and N50 of 6.65 Mb. They annotated the genome where they report 23,923 genes. They suggest with parsimonious analysis that gene family such as plant cell wall degrading enzymes; aspartyl protease domain containing genes are expanded. The author finds reduction in genes responsible in catabolism of toxins. They report the first whole-genome sequence from the genus Ips and allude to the importance of this genome to tackle evolutionary and ecology questions in this pest as well as design genetic manipulation experiment for its eradication. The manuscript read well, but I have few minor comments on the manuscript that need to be addressed:	
1)- line 133. The author used a kmer approach to estimate genome size, which is GenomeScope 2.0 relying on jellyfish analysis. First, is there a flowcytometry measure for Ips typographus and how your kmer estimate will compare to that? Second, using their kmer approach, if it is possible to mention which kmer they use for this analysis. It would be nice if they run multiple kmers and show some GenomeScope figure results, with heterozygosity estimates. Third, in case there is no flowcytometry measure available, it would be great to have the author run another tool such as KmerFreq_AR in SOAPdenovo2 at similar kmers to see if they get the same results as GenomeScope.	We have now included the results of flow cytometry genome size estimation provided to us from personal communication (Line 226-230). The estimated size calculated using this method was similar to the size estimates made using the k-mer counting method (Line 229).
2)- Line 223. The author report 28.2 % of the Ips typographus made of repeats. They used Repeatmasker, which will generate a report for the different classes of repeats (Long terminal repeats (LTR), DNA etc.). Since it is about the genome, and comparative genomics. It is better to report the different classes of TEs from RepeatMasker and do the comparison with other published genomes reported in Supplementary Table 2. Is there any correlation between genome size and TEs?	A breakdown of different classes of repeats annotated in the I. typographus genome has now been included in Supplementary Table 8, as suggested.
3)- The author discuss in Line 283 and 298, which GH48 is part of the Ips typographus genome, but also they talk about horizontal gene transfer from bacteria or fungi. In addition, they cite that gut microbiome of Ips typographus is the place to help the species cope with its enzymatic digestion during colonization. I would like to ask if the author did any analysis to rule out its horizontal transfer origin from fungi? They have mentioned	The phylogenetic analysis was performed with only the protein sequences obtained from the genome annotations of the species used in this study to highlight the homology with other coleopterans. There are of course matches to GH48 protein sequences in the publicly available fungal databases, however, the hits from all eight GH48 proteins found in Ips have far higher levels of similarity to those

phylogentic analysis of GH48, which has similarity to other Scolytinase species D. ponderosae and H. hampei. Did they search similarly to Fungi database?	found in beetles than to any other organism.
4)- The author did comparative genomics and Phylogenomics line 161....171 and report expansion and contraction of gene families using a parsimonious approach and Fisher exact test. Usually, statistical significance of the family size changes should be inferred across a species phylogeny using a stochastic birth-death process based model such as the one implemented in CAFÉ from De Bie et al. 2006. I was wondering if the authors can run their analysis using this tool, would they expect the same results for the expanded and contracted gene families as they have reported? The authors have to address these points, and then it will be considered for publication.	The CAFE analysis compares genes clustered into groups based on orthology whereas we had compared genes based on functional domain motifs. These are quite different approaches, and one would expect different outcomes. We have now performed the CAFE analysis, and found that this approach determined far more gene groups to be expanded (1,065), containing almost a third of all the predicted genes. Contained within these genes were those we had determined to be significantly expanded using Pfam domain analysis and we chose to focus on these Pfam results given the functional aspect of the domain analysis approach. The CAFE analysis results are described on Lines 289-294.
Reviewer #4:	
This paper reported the genome of the Eurasian spruce bark beetle (Ips typographus), which is one tree-killing species and causes economic and ecological costs. Decoding the genome of this beetle is helpful to develop novel pest control strategies to mitigate its outbreaks in forest. However, to get results with higher quality and thus provide deeper insights into this species, some analysis strategies should be changed and more works should be done. Questions and problems are listed below.	We thank you for the assessment of the manuscript and the many valuable suggestions for improvements.
1. The results of completeness evaluation of this draft genome through BUSCO tool were not consistent in this manuscript. From the line 225 to 227, the author stated that “1,649 (99.5%) of the 1,658 genes in the insecta_odb9 database could be identified as present either partial or complete”, while from the line 241 to 244, the author stated that “searches of the annotated predicted protein sequences using BUSCO from the insecta_odb9 libraries identified 1,584 (95.5%) of the 1,658 protein sequences in the Insect dataset”. What induced this difference, and what is the correct one?	The two sections of the manuscript referred to here are separate analyses. The first is a search performed on the genome, the second is a search performed on the gene models, thus they are both correct and are not expected to be numerically identical. The searches produce different results due to the gene modelling not capturing a small percentage of the set of BUSCOs found in the genome. We understand this to be a limitation. However, manually curating all of the gene models is outside the scope of this study. We describe the two different BUSCO analyses on lines 128-130 and 186-188.
2. From the line 250 to 252: “an overall average of 93.6% mapped reads indicates that the majority of transcripts captured in our RNA-seq data can be	This was a poor word choice. The word “intact” has been changed to “represented” on line 269 in the revised manuscript.

found and are intact in the I. typographus gene set”. Only mapping rate of RNA-seq data can’t indicate the intact structure of gene models, thus this statement is not very reasonable.	
3. From the line 262 to 264, the author performed a gene ontology enrichment analysis for the unique set of 811 gene clusters in I. typographus and found some enriched functions. First, the method adopted in this analysis should be clarified. Second, these enriched functions seem very general and important for any animals. Thus, more details should be listed and discussed to reveal the uniqueness of these enriched functions; otherwise it is tended to suspect there are some problems in the identification of unique set of 811 gene clusters in I. typographus due to technique errors.	We do not agree with the argument that as the enriched gene ontology terms were found to be somewhat general in nature that the identification of these gene clusters must be due to technical error. Our analysis is done according to standard methods and the terms quite clearly relate to DNA processing mechanisms.
4. From the line 264 to 266, the author identified 22 gene families which were significantly expanded in I. typographus when compared with the 11 other coleopteran genomes and a selection of other model invertebrates. Expanded gene families in I. typographus were identified taking the average member counts of these gene families in remaining species as the background through Fisher’s exact test. I challenge that the average member counts of gene families in the remaining species shouldn’t be considered as the background, because the average statistical value is misleading in this context. Expansion or contraction of gene families in a species is a relative status compare to the most recent common ancestor (MRCA), whereas the average member counts of gene families in remaining species can’t represent their ancestral status. The CAFE software package should be used to identify expanded gene families.	Our protein domain-based approach for revealing expanded gene families has been described previously (for example please see Hall et al, 2017, Nature; Albertin et al, 2015, Nature; Simakov et al, 2012, Nature). However, we also performed the CAFE analysis which resulted in a far greater number of expanded gene families (1,065) and have included this in the manuscript as described above. Genes from our protein domain analysis were among the genes identified as expanded from the CAFE analysis. We believe the focus on gene function is of greater interest and have reported our results accordingly. The new CAFE analysis results are described on Lines 289-294.
5. From the line 267 to 269, the author stated that “the limited number of significantly expanded gene families discovered in I. typographus in this analysis reflects the comprehensive array of beetle species included in the dataset”. This logic is hard to follow, because the number of expanded or contracted gene families is related to the evolutionary pattern but not the sampling strategy.	We acknowledge that this sentence was confusing, and it has been removed from the revised manuscript.
6. From the line 275 to 277, the author identified significantly expanded gene families associated with plant cell wall degrading enzymes in I. typographus. Similar to the problem I mentioned above, expanded gene families should be identified by the CAFE software package.	In the response above we describe the precedent for using functional protein domains for revealing gene family expansions. As requested, we have performed the recommended CAFE analysis (explained above).
7. From the line 286 to 288, the author described	We show the expression pattern for each

the surprised expressed pattern of glycosyl hydrolase (GH) enzymes in I. typographus (which is predominant in the fat body), but it failed to provide more detailed results and citations to support or explain this phenomenon.	GH48 gene across 13 different life stages and tissues - we believe the level of detail we provide is adequate to report this finding. This, however, is a novel result and consequently, there is very little data in the literature to help explain this phenomenon and its biological relevance. Thus, we have stated that further studies are required to investigate functional roles for these proteins (Lines 309-311) . Such studies are outside of the scope of this report, and we wish to not speculate too much around this finding (due to the lack of literature that could help explaining the outcome).
8. From the line 297 to 300, the author found “three separate tandem duplication events appear to have occurred in I. typographus since the establishment of GH48 genes in the Curculionoidea, seen as three pairs of genes occurring adjacent to each other in the genome (sequentially numbered gene models are found adjacently in the genome)”. First, the physical distance and sequence similarity between these three pairs of genes should be displayed with a table or a figure. Genes generated through tandem duplication events are close to each other in the physical distance and similar to each other in the gene sequence. Only sequential number of these gene models is not enough to indicate tandem duplication events. Second, a synteny plotting among species in the Curculionoidea should be provided to reveal whether the tandem duplication events of this gene happened in other related species. Otherwise, the conclusion of “three separate tandem duplication events appear to have occurred in I. typographus since the establishment of GH48 genes in the Curculionoidea” can’t be made.	We had made the statement “sequentially numbered gene models are found adjacently in the genome” simply to help the reader identify the genes. We inspected the genome location to precisely identify proximity, though we understand this may be confusing for the reader. We have removed this sentence from the main text. Synteny plotting is informative when performed with closely related species, however, our comparisons are with species that are quite divergent, and such analysis is outside the scope of this study.
9. The citation information is not complete in the line 316.	Here (line 336) we add a note to the in-text citation that we also refer to references within the paper cited. We understand this to be the accepted convention.
10. From the line 323 to 325, the author stated “these two conifer-infesting bark beetles did not display an enrichment for gene families involved in xenobiotic metabolism when compared to other agriculturally important Coleopterans”. This conclusion can’t be made without any evidences got from statistical test. Although the member counts of these gene families in I. typographus are smaller than other species, the background of	We acknowledge that this was poorly phrased. We have changed the structure of this paragraph beginning on Line 343. There is an implicit Hypotheses (H): “Bark beetles should have stronger defense to toxic plant chemicals”), which the referee has caught and reacted on as any statistical analysis of data (which actually clearly negates the H), testing the hypotheses is not

other gene families, the quality of genome assemble as well as gene annotation and whether the difference is significant should be considered.	given. This text is now fully revised (lines 343-350), including a H and simple test thereof, using Effect Size. The effect size calculation is also explained in the Materials and Methods (Lines 215-218).
11. From the line 329 to 332, the author stated “it is possible that Scolytinae specialists on conifers detoxify host defences (such as abundant conifer monoterpenes and phenolics) using a smaller, but specialized, repertoire of P450 enzymes, as compared to generalist beetles feeding on diverse angiosperms”. And from the line 335 to 337, the author suggested “this may appear paradoxical considering its lifestyle, but I. typographus may have a strategy of quickly reducing the levels of tree defences by mass-attack and assistance from fungal symbionts”. Both of these two statements are built on the conclusion that there is no enrichment in I. typographus for gene families involved in xenobiotic metabolism. As is mentioned above, this conclusion is not solid. Further, to make a whole research, some experiments associated with functional genomics should be done, such as cloning some key genes coding P450 enzymes and testing their activity. In addition, 16S rDNA in the gut of I. typographus should be sequenced to reveal the pattern of its fungal symbionts.	We believe that the functional genomics study is well outside of the scope of this study. Complementary studies are published by some of the co-authors of this paper, investigating the gut symbionts for both their bacterial and fungal components, using ‘omics’ diversity profiling techniques. They cover the overall aspects asked for, as far as we can see. Chakraborty, A., M. Z. Ashraf, R. Modlinger, J. Synek, F. Schlyter, and A. Roy. 2020. Unravelling the gut bacteriome of Ips (Coleoptera: Curculionidae: Scolytinae): identifying core bacterial assemblage and their ecological relevance. Scientific Reports 10 (1): 18572. Chakraborty, A., R. Modlinger, M. Z. Ashraf, J. Synek, F. Schlyter, and A. Roy. 2020. Core mycobiome and their ecological relevance in the gut of five Ips bark beetles (Coleoptera: Curculionidae: Scolytinae). Frontiers in Microbiology 11:2134. We feel that there is sufficient evidence in the literature to include this concept in our discussion and that we are basing this from clear differences in gene family numbers. Also, we would like to point out that the text on original lines 329-337 did not state any “conclusions” but was rather meant as a discussion. We use quite careful wording here (i.e., “this may appear paradoxical”, and “I. typographus may have a strategy”...), and some speculation should be allowed, when properly worded.
12. From the line 352 to 355, the author concluded that “the essential odorant receptors are one example of gene family that could be targeted to limit outbreaks, since interfering with the beetles’ odour-guided host and mate finding behaviours likely will reduce their reproductive success”. However, there is no any related analysis been done in this manuscript, and it should be added to make deeper insights into this species I. typographus.	We believe this to be out of scope for this paper and is only stated for illustrative purposes explaining the utility of the genome and its application with an important area of research. It would be a large study in itself to perform such an analysis. We cite two papers that have performed analyses of the odorant receptors based on transcriptome data, and one of them was published recently (Yuvaraj et al., 2021) (Line 381).
13. From the line 143 to 149, the author adopted the transcriptome de novo assembled from the RNA-Seq data of I. typographus through Trinity as evidence for homology-based gene prediction. In this way, assembled errors might be taken into the	We specifically state here that due to the potential for misassembly, transcriptome data were only used as evidence and not for prediction. Reviewer #4 has misread this sentence.

model used for gene prediction. Thus, it is more appropriate to use RNA-Seq data only as transcript-based evidence.	
14. The line 178: “orthologous gene groups were used to build a rooted species tree”. What kind of orthologous gene groups were adopted to build the rooted species tree? In fact, single-copy orthologous gene groups should be used.	Yes, we used single copy orthologs to build a new rooted species tree. We have updated the methods in this section to reflect this (Lines 208-210).
15. From the line 185 to 187, the author prepared transcriptome libraries for callow male gut and callow female gut of I. typographus . It should be clarified that how to distinguish the transcriptomes of I. typographus and its gut symbiont.	Prokaryote mRNA is very unstable and presents a tiny amount considering the more stable eukaryotic host mRNA with poly A tail. We used the Poly A enrichment method during library preparation to enrich the host mRNA. Hence, the amount of mRNA from the symbionts after RNA extraction should be negligible.
16. The line 198: the version of Trimmomatic used in this analysis should be clarified.	The missing version number has now been included (Line 155).

REVIEWERS' COMMENTS:

Reviewer #1 (Remarks to the Author):

This manuscript had four fairly lengthy reviews and the authors have addressed the concerns. I have no further issues with the manuscript and recommend it be published.

Reviewer #3 (Remarks to the Author):

The authors address the different points as well as they improved the manuscript quite well. I would certainly recommend it for publication.
Thank you